# Sources of Health Anxiety for Hospital Staff Working during the Covid-19 Pandemic

**DOI:** 10.3390/ijerph18063094

**Published:** 2021-03-17

**Authors:** Mehran Shayganfard, Fateme Mahdavi, Mohammad Haghighi, Dena Sadeghi-Bahmani, Serge Brand

**Affiliations:** 1Department of Psychiatry, Arak University of Medical Sciences, Arak 3848176341, Iran; mshayganfard@arakmu.ac.ir; 2Endocrinology and Metabolism Research Center, Arak University of Medical Sciences, Arak 3848176341, Iran; 3Student Research Committee, Arak University of Medical Sciences, Arak 3848176341, Iran; fatememahdavi111@gmail.com; 4Research Center for Behavioral Disorders and Substances Abuse, Hamadan University of Medical Sciences, Hamadan 65174, Iran; dr_haghighi_ps@yahoo.com; 5Departments of Physical Therapy, University of Alabama at Birmingham, Birmingham, AL 35209, USA; dena.sadeghibahmani@upk.ch; 6Sleep Disorders Research Center, Kermanshah University of Medical Sciences, Kermanshah 67146, Iran; 7Center for Affective, Stress and Sleep Disorders (ZASS), Psychiatric University Hospital Basel, 4002 Basel, Switzerland; 8Department of Clinical Research, University of Basel, 4031 Basel, Switzerland; 9Division of Sport Science and Psychosocial Health, Department of Sport, Exercise and Health, University of Basel, 4052 Basel, Switzerland; 10Substance Abuse Prevention Research Center Health, Kermanshah University of Medical Sciences, Kermanshah 67146, Iran; 11School of Medicine, Tehran University of Medical Sciences, Tehran 25529, Iran

**Keywords:** health anxiety, state-anxiety, trait-anxiety, COVID-19, frontline hospital staff members, age, gender

## Abstract

Background: During the COVID-19 pandemic, the likelihood that hospital staff will report symptoms of depression, anxiety, and stress has increased. The aim of this study was to evaluate the relative influences of circumstantial, demographic, and trait–state anxiety variables on health anxiety in this group. Methods: A total of 168 hospital staff members (mean age: 28.91 years; 56.5% females) participated in the study. They completed a series of questionnaires covering sociodemographic characteristics, health anxiety, state–trait anxiety, and job-related information. Participants also reported whether they had close acquaintances (friends, family members) infected with COVID-19. Results: Higher health anxiety was related to both trait and state anxiety. Working on the frontline, being in contact with close acquaintances infected with COVID-19, and higher state and trait anxiety predicted higher health anxiety. Gender, age, and educational background were not predictors. Conclusions: In a sample of hospital staff, subjective feelings of anxiety about one own’s health were related to personality traits, individual experiences of having close acquaintances infected with COVID-19, and working on the frontline.

## 1. Introduction

As are all countries, Iran is likewise struggling with the COVID-19 pandemic. Dealing with the COVID-19 pandemic is a challenge for societies, individuals, and healthcare systems [1,2]. The national government, advised by health authorities, has imposed restrictions on movement to reduce the risk of spreading the virus and to limit severe cases of infection and additional deaths [3,4]. To this end, state authorities have temporarily legislated to close borders, schools, shops, markets, services, universities, sports events, and religious and cultural centers, and to forbid gatherings in public. By 14 January 2021, and following the COVID-19 health data (https://covid19.healthdata.org/iran-(islamic-republic-of)?view=resource-use&tab=trend&resource=all_resources) (accessed on 20 January 2021), Iran needed 16,291 hospital beds per day to treat COVID-19 patients, while about 45,159 beds were available. It follows that as regards capacities to treat COVID-19 patients, sufficient resources were available. 

However, these positive numbers for available hospital beds should be balanced against the cost in human resources. Data from rapid reviews and meta-analyses have shown that hospital staff members in contact with infected patients (“frontline hospital staff members”) appear to be at increased risk of reporting symptoms of exhaustion, depression, anxiety [5], short- and long-term mental health problems [6], acute and post-traumatic stress [7,8,9], and psychological distress [5,10]. Batra et al. [11], in their meta-analysis, summarized results from 65 studies, including 79,437 hospital staff members, and they reported the following prevalence rates: anxiety: 34.4%; depression: 31.8%; stress: 40.3%; post-traumatic stress syndrome: 11.4%; insomnia: 27.8%; psychological distress: 46.1%, and burnout: 37.4%. The following specific risk factors were identified: lower age, being more junior, being the parents of young children, having an infected family member, lack of practical support, stigma [5,7], heavy workload, lack of training, lack of social support, and limited work experience [7]. Depression, anxiety, and psychological distress were also common among hospital staff members in contact with infected patients [12].

As regards the psychological health of Iranian hospital staff members, higher scores for job-related stress and burnout were related to working on the frontline [13], irrespective of age, job experience, or the social support of friend and families [14]. In contrast, among frontline nurses with an average age of 40.6, higher age, higher educational degrees, and being male predicted lower stress [15,16]. A higher task load was associated with lower general health [16]. In addition, frontline nurses reported anxieties related to the disease, fear of infecting their families, emotional distress related to delivering bad news, and conflicts between (dysfunctional) fears and the need to discharge a vital job [17]. Of 761 nurses surveyed in one study, 267 reported fear of getting infected [18]. 

To summarize, while there is extant evidence that during the COVID-19 pandemic, both Iranian and non-Iranian hospital staff members have been more likely to report symptoms of depression, anxiety, post-traumatic stress [7], and secondary trauma [8], to our knowledge, there has so far been no study of health anxiety. Health anxiety is the subjective fear that one own’s health might be in danger [19]. We decided to assess health anxiety and not just state or trait anxiety for the following reasons. First, unlike state or trait anxiety, health anxiety concerns the cognitive–emotional belief that one’s health is in danger, and in this respect, it is considered a health belief model. Studies on health belief models [20,21,22] have shown that subjective beliefs impact a person’s susceptibility to illnesses. Second, health belief models, and in this case health anxiety, influence the degree to which a person believes she/he is more or less susceptible to infection by the COVID-19 virus. Third, health belief models share a common basis with the cognitive–emotional stress model of Lazarus and Folkman [23] and research on placebo effects [24,25,26]. It is not the COVID-19 virus per se [27], or stress per se [23], or a placebo per se [25,26,28] that produces an emotional, cognitive, and behavioral effect, but the subjective and cognitive–emotional meaning projected onto the virus, to a work environment, or to a placebo. Fourth, given this background, the present study differs from previous studies in focusing on the subjective and cognitive–emotional and behavioral responses of healthcare workers during the pandemic, and not on objective realities. Fifth, from the above, it follows that purely individual cognitive–emotional processes impact on perceptions of danger, which here is the subjective risk of getting infected by the virus. In a previous study of women during the peripartal stage, health anxiety and being close to people infected with COVID-19 were the main drivers of postponement or avoidance of routine medical appointments [27]. Thus, one might expect health anxiety to be related to subjective but not objective exposure to danger in general and to COVID-19 specifically. Given this background and given the lack of data on this topic among hospital staff in Iran, the key aim of the present study was to investigate the associations between health anxiety and other symptoms of anxiety (state vs. trait anxiety), being exposed to close acquaintances infected with the virus and work-related circumstances. To this end, 168 hospital staff completed a series of questionnaires on health anxiety, state and trait anxiety, educational background, and exposure to others infected with COVID-19 in their private lives. 

The following two hypotheses and one research question were formulated. First, previous studies showed that being close to people infected with COVID-19 would increase psychological distress, higher anxiety, and avoidance behavior [17,18,27]; accordingly, we expected that being close to people infected with COVID-19 would be associated with higher health anxiety. Second, others have shown that hospital staff members working on the frontline reported higher symptoms of distress and anxiety [12,13]; given this, we anticipated that working on the frontline would be associated with higher scores for health anxiety and trait and state anxiety. Next, we explored which of the sociodemographic and work-related characteristics, state–trait anxiety, and being close to people infected with COVID-19 would independently predict health anxiety. 

We believe that findings from this study have the potential to identify those hospital staff at greatest risk of reporting health anxiety and thus in need of extra support in coping with COVID-19-related psychological issues. 

## 2. Methods

### 2.1. Procedure

Hospital staff at the Arak University of Medical Sciences (Arak, Iran) and Hamedan University of Medical Sciences (Hamedan, Iran) were approached from May to July 2020 to participate in a cross-sectional study on dimensions of anxiety during the COVID-19 pandemic. To this end, the study was posted on the intranet and social network sites of the two hospitals. Staff members interested in participation could click and follow the link to be contacted by one of the study team. All participants were informed about the aims of the study and that their data would be anonymous and held securely. Thereafter, they signed a written informed consent. Participants completed a series of questionnaires covering sociodemographic, workplace-related, and anxiety-related matters (see below). The ethical committee of the Arak University of Medical Sciences (ARAKMU; Arak, Iran) approved the study (approval no: IR.ARAKMU.REC.1399.014), which was performed in accordance with the seventh and current edition [29] of the Declaration of Helsinki.

### 2.2. Participants

Of 180 individuals who initially responded, 168 (93.3%) agreed to participate in the survey. Inclusion criteria were: (1) Age of 18 years and older; (2) Hospital staff member of the Arak and Hamedan University of Medical Sciences; (3) Willing and able to comply with the study conditions; (4) Working full-time; and (5) Signed written informed consent. 

### 2.3. Measures

Sociodemographic and workplace-related information: participants reported their age (in years), gender (male, female), civil status (single, married, divorced, widowed), number of children, current job position (post graduate medical doctors; medical students; nursing staff), whether working frontline caring for patients with COVID-19 (yes vs. no), and being in close contact with a person infected with COVID-19 (e.g., a friend, a family member; answers: “no”, “yes, but I don’t have any close contact with him/her”; “yes, and I do have a close contact with him/her”).

### 2.4. State and Trait Anxiety

Participants completed the State–Trait Anxiety Inventory (STAI) [30]. Psychometric properties of the Farsi version for adults [31] and adolescents [32] were satisfactory. The STAI consists of 42 items. Typical items for an anxiety state are “I feel relaxed”; “I feel nervous”, and “I feel tense”. Typical items for anxiety traits are “I get nervous and restless when thinking of all my duties and issues” and “I can’t stop ruminating about unimportant stuff”. Answers were given on an 8-point rating scale with the anchor points 0 (not true at all) and 7 (completely true), and with higher sum scores reflecting higher state and trait of anxiety (Cronbach’s alphas; trait anxiety: 0.87; state anxiety: 0.88).

### 2.5. Health Anxiety Questionnaire 

As already employed elsewhere [27], participants completed the Persian version [33] of the Health Anxiety Inventory [19]. The questionnaire consists of 18 questions focusing on anxiety about one own’s health, ranging from no fear concerning health at all to hypochondriasis (DSM-IV; [34]) or somatic symptom disorder and illness anxiety disorder (DSM-5; [35]). Hypochondriasis is understood as the clinical and dysfunctional belief that one is suffering from or going to be infected by a severe and dangerous disease. Typical items are as follows: (1) (a) “I do not worry about my health”; (b) “I occasionally worry about body vigilance and my health”; (c) “I spend much of my time worrying about my health”; (d) “I spend most of my time worrying about my health”; (2) (a) ”I notice aches/pains less than most other people (of my age)”; (b) “I notice aches/pains as much as most other people (of my age)”; (c) “I notice aches/pains more than most other people (of my age)”; (d) “I am aware of aches/pains in my body all the time”; (3) (a) “As a rule, I am not aware of bodily sensations or changes”; (b) “Sometimes I am aware of bodily sensations or changes”; (c) “I am often aware of bodily sensations or changes”; (d) “I am constantly aware of bodily sensations or changes”. Items are aggregated to the following dimensions: illness severity, illness likelihood, body vigilance, and total score. Higher sum scores reflect more pronounced health anxiety (Cronbach’s alpha: 0.87).

### 2.6. Statistical Analysis 

All 168 participants fully completed all questionnaires; no data had to be discarded, and no missing data had to be replaced.

Correlations (Pearson’s and Spearman’s rank correlations) were computed for associations between age, state and trait anxiety, health anxiety, and being close to people infected with COVID-19. 

A series of ANOVAs was performed with the following factors: being close to people infected with COVID-19 (yes, and close contact; yes, but no close contact; no) and with working on the frontline (yes vs. no), and state and trait anxiety and health anxiety as dependent variables. Cut-off values for partial eta-squared were $\eta_{p}^{2}$ < 0.019 = trivial effect size (T); 0.02 $<\eta_{p}^{2}$ < 0.059 = small effect size (S); 0.06 < $\eta_{p}^{2}$ < 0.139 = medium effect size (M); $\eta_{p}^{2}$ > 0.14 = large effect size (L) ((ES) = effect size).

A multiple regression analysis was performed to calculate whether age, gender, educational background, state anxiety, trait anxiety, working on the frontline, or being close to people infected with COVID-19 predicted health anxiety. Following others [36,37], preliminary conditions to perform a multiple regression analysis were met: the sample size was >100; the number of predictors × 10 should not be greater than sample size (here: 8 × 10 = 80 < 168); predictors should sufficiently explain the dependent variable (R = 0.689; R^2^ = 0.469), and the Durbin–Watson coefficient should be between 1.5 and 2.5, indicating that the residuals of the predictors were independent of each other.

The level of significance was set at alpha = 0.05.

All computations were performed with SPSS^®®^ 25.0 (IBM Corporation, Armonk, NY, USA) for Apple Mac^®®^. 

## 3. Results 

### 3.1. Participant Characteristics

Table 1 provides an overview of participants’ sociodemographic and work-related characteristics, both for the whole sample and separately for males and females. 

Briefly, 168 participants were included in the study. Of these, 73 (56.5%) were females. Participants’ mean age was 28.91 years (SD = 0.6.63). In addition, 47 (28%) were working on the frontline. A total of 19 (11.3%) had no contact with people infected with COVID-19; 104 (61.9%) had a close acquaintance infected with the virus, but there was no contact, and 45 (26.8%) had a close acquaintance who was infected and they were in close contact with this person. 

Table 2 provides descriptive and correlational statistical indices for age, state and trait anxiety, health anxiety, and being close to people infected with COVID-19. 

Age was unrelated to state or trait anxiety or to health anxiety. Higher age was associated with being less or not at all in contact with close people infected with COVID-19.

State and trait anxiety and health anxiety were highly interrelated. 

Being close to people infected with COVID-19 was related to higher state and trait anxiety and to health anxiety.

### 3.2. Being Close to People Infected with COVID-19 (First Hypothesis) and Frontline Working (Second Hypothesis)

Table 3 and Table 4 provide descriptive and inferential statistical indices for state and trait anxiety and health anxiety as dependent variables and being close to people infected with COVID-19 and frontline working as factors.

State and trait anxiety and health anxiety were significantly higher in participants who reported being close to people infected with COVID-19 (medium to large effect sizes). Trait anxiety and anxiety total score were significantly higher in those participants working on the frontline (p’s < 0.01; small to medium effect sizes). Frontline working had no impact on health anxiety. The interactions were non-significant (trivial effect sizes). 

### 3.3. Predicting Health Anxiety (Research Question)

A multiple regression analysis was performed to determine whether age, gender, job status, state anxiety, trait anxiety, frontline working, or being close to people infected with COVID-19 predicted health anxiety. Table 5 provides all statistical indices.

Higher state anxiety, higher trait anxiety, being in contact with a close acquaintance infected of COVID-19, and frontline working predicted higher scores for health anxiety. Age, gender, and current job status were excluded from the equation, as these variables did not reach statistical significance.

## 4. Discussion

The key findings of the present study were that among a sample of hospital staff working during the COVID-19 pandemic, several factors independently predicted health anxiety; these factors were higher trait anxiety, which was a personality trait predisposition, higher state anxiety, being frontline hospital staff, and being in direct contact with close acquaintances infected with COVID-19. In contrast, age, gender, and current educational background were unrelated to health anxiety. The present results add to the current literature in showing that among hospital staff members working during the COVID-19 pandemic, scores for dysfunctional fears about health tending toward hypochondriasis (sensu DSM-IV [34]), or toward somatic symptom disorder and illness anxiety disorder (sensu DSM-5 [35]) were related to personality traits and being exposed both in their private lives and at work to people infected with COVID-19. To put it more simply, fears about health reflected both personal and job-related factors. 

Two hypotheses and one research question were formulated, and each of these is considered in turn.

Our first hypothesis was that those who were close to others infected with COVID-19 also reported higher anxiety about their health, and this was confirmed. It follows that the current results are in accord with previous findings [17,18,27]. However, we expanded upon previous research in showing that personal circumstances were completely independent of the work-related context in their impact on health anxiety among hospital staff during the COVID-19 pandemic. The pattern of results also suggests that at the individual level, thorough counseling and preventive strategies at work might not protect against health anxiety. Rather, personal circumstances such as being close to people infected with COVID-19, which again was related to the dysfunctional belief that one’s own health was at risk, appeared to be more powerful for psychological functioning. 

Our second hypothesis was that frontline working with patients infected with COVID-19 would be associated with higher scores for health anxiety and trait and state anxiety, but this was only partially confirmed. When introducing frontline working as an independent predictor in the regression model, this had an impact on health anxiety (see Table 5). However, when introducing frontline working as an independent factor alongside being close to people infected to COVID-19, then the statistical impact of the former on health anxiety was reduced (Table 3 and Table 4). In contrast, working frontline was related to higher trait anxiety. Overall, the present data only partially confirmed previous findings [12,13]. 

Interestingly, working on the frontline was associated with trait but not with state anxiety. In other words, it was related to the personality trait of anxiety but not to current and transient exposure to possible danger and thus to state anxiety. The data available to us cannot clarify why this pattern of results was found. It would appear to imply that hospital staff high in trait anxiety prefer working on the frontline, and that frontline-related state anxiety is a consequence but not a cause of health anxiety, although this seems highly unlikely. Rather, we believe this was probably a random result, which at this stage is in need of thorough replication and theoretical elaboration. 

Our research question explored whether sociodemographic or work-related factors, state–trait anxiety, or being close to people infected with COVID-19 predict health anxiety. The pattern of findings suggested that health anxiety was the complex result of personal concerns [5], contact with close acquaintances infected with COVID-19 [5], and working on the frontline [13,17], while age, educational background, and gender were unrelated. Such lack of effects have been reported previously [14], though for age [5,15], educational background [5,15], and gender [15,16], opposite findings have also been reported. 

Next, a key question is how to treat health anxiety prospectively among hospital staff working with COVID-19 patients. As mentioned in the Introduction, health anxiety is the dysfunctional belief that one’s health is in danger [19]. Given the discussion there of health anxiety as a health belief model, it follows that merely changing the environment or increasing safety and security conditions in the hospital must by definition fail. Online cognitive–behavioral therapy interventions (CBT) have been shown to be no less effective than conventional face-to-face CBT for individuals with health anxiety [24]. Our view is that only CBT interventions have proven to offer efficacy, precision, economy, reliability, and success in treating health anxiety. 

Despite the novelty of the results, the following limitations should be considered. First, by default, it was impossible to review and retrieve all of the relevant and rapidly growing literature on the topic. To illustrate, in PubMed^®®^, the term “COVID-19” yielded 93,438 hits (retrieved 18 January 2021), “COVID-19 and psychiatry” yielded 3694 hits; “COVID-19 and hospital staff” yielded 2558 hits; “COVID-19 and staff members and Iran” yielded 66 hits, “COVID-19 and staff members and review” yielded 680 hits, “COVID-19 and staff members and meta-analysis” yielded 26 hits; “COVID-19 and staff members and anxiety” yielded 312 hits. A recent umbrella review summarized the results of seven current systematic reviews and meta-analyses [38]. Relatedly, other studies and a narrative review have investigated associations between psychological issues of healthcare workers during the pandemic and post-traumatic stress symptoms (PTSS [7]), post-traumatic stress disorders (PTSD; [9]), and secondary trauma [8]. These publications reported that compared to healthcare workers not on the frontline, frontline workers had higher incidence rates for symptoms of anxiety, depression, burnout, and stress. Given this background, we acknowledge that the literature reviewed in the Introduction and the range of psychological issues considered will necessarily be incomplete. Second, the survey was performed anonymously, although only participants willing and able and to comply with the study conditions were enrolled. Consequently, the sample, which also was relatively small, may not be fully representative of hospital staff. Given this, sample characteristics might have biased the pattern of results. Relatedly, third, it is conceivable that latent and unassessed dimensions such as leisure time activities, physical activity patterns, substance use, social support, medication use, sleep hygiene-related behavior, and pre-existing vulnerabilities might have biased two or more dimensions in the same or opposite directions. Fourth, by definition, multiple regression models imply a theoretical framework of causes and consequences; in contrast, and strictly speaking, a cross-sectional study design cannot support causal inferences. However, a close consideration of the variables that independently predicted health anxiety (see Table 5) suggest that state and trait anxiety, frontline working, and living in proximity to people infected with COVID-19 caused health anxiety, while influences in the opposite direction (e.g., health anxiety leads to working on the frontline and to living with people infected with COVID-19) are highly unlikely. Given this, a follow-up study would have allowed us to investigate the causality of these relationships more thoroughly. Fifth and similarly, future interventional studies should investigate what workplace-related preventive measures can reduce health fears among hospital staff involved in the treatment of patients with COVID-19. To take one example, there is evidence that frontline nurses appreciate supportive resources and social support [39]. Furthermore, among 599 healthcare practitioners working in eight different Iranian cities, quality of life was associated with higher social support [40].

## 5. Conclusions

Among full-time hospital staff working during the COVID-19 pandemic, a combination of personal circumstances and job-related conditions predicted health anxiety, while age, gender, and educational level were unrelated. Given this, it appears that interventions to avoid health anxiety should be focused on both individual and workplace-related levels. However, hospital management has very little scope to influence staff members’ personality traits and so should seek to limit the amount their staff work on the frontline so as to reduce their anxieties about health. At a personal level, short-term cognitive–behavioral interventions should help hospital staff estimate risks of infection more appropriately.

## Figures and Tables

**Table 1 ijerph-18-03094-t001:** The overview of sociodemographic and questionnaire-related descriptive data.

Variable		Total N (%)	Male N (%)	Female N (%)
Civil status	Single	84 (50)	35 (47.9)	49 (51.6)
	Married	82 (48.8)	36 (49.3)	46 (48.4)
	Divorced/Separated Widowed	2 (1.2) 0 (0)	2 (2.7) 0 (0)	0 (0) 0 (0)
	total		73 (100)	95 (100)
Educational background	Post Graduate Medical Doctors	20 (11.9)	11 (15.1)	9 (9.5)
	Medical Students	52 (31)	24 (32.9)	28 (29.5)
	Nursing Staff	96 (57.1)	38 (52.1)	58 (61.1)
	total		73 (100)	95 (100)
Working frontline	Yes	47 (28)	22 (30.1)	25 (26.3)
	No	121 (72)	51 (69.9)	70 (73.7)
	total		73 (100)	95 (100)
Being close to a person infected with COVID (no/yes, but not close; yes and close)	No Yes, but not close Yes, and close total	19 (11.3) 104 (61.9) 45 (26.8)	4 (5.5) 43 (58.9) 26 (35.6) 73 (100)	15 (15.8) 61 (64.2) 19 (20) 95 (100)
	M (SD)		M (SD)	M (SD)
Age (year)	28.91 (6.62)		30.26 (7.52)	27.87 (5.661)
Health anxiety	Illness Likelihood Illness Severity Body Vigilance	11.09 (4.346) 4.2 (1.972) 4.52 (1.821)	11.4 (4.618) 4.21 (2.108) 4.55 (1.864)	10.85 (4.133) 4.2 (1.871) 4.49 (1.798)
Health anxiety Total		19.73 (7.307)	20.10 (7.672)	19.44 (7.042)
State–trait anxiety	State anxiety	42.81 (9.117)	43.66 (9.449)	42.16 (8.848)
	Trait anxiety	41.1 (8.403)	42.22 (8.999)	40.24 (7.855)
State–trait anxiety Total		83.91 (16.384)	85.88 (17.41)	82.4 (15.474)

General overview: Correlations between age, state and trait anxiety, health anxiety, and being close to people infected with COVID-19.

**Table 2 ijerph-18-03094-t002:** Descriptive and correlational statistical indices (Pearson’s correlations) of age, state and trait anxiety, health anxiety, and being close to people infected with COVID-19 (Spearman rank correlations).

Dimensions	State-Trait Anxiety			Health Anxiety				Being Close to People Infected with COVID-19	Descriptive Statistics
	State	Trait	Total Score	Illness Likelihood	Illness Severity	Body Vigilance	Total Score	---	M (SD)
Age (years)	0.07	0.10	0.09	0.12	0.08	0.06	0.11	−0.29 ***	28.91 (6.62)
State–trait anxiety									
State anxiety	−	0.75 ***	0.94 ***	0.58 ***	0.47 ***	0.41 ***	0.58 ***	−0.29 ***	42.81 (9.12)
Trait anxiety		−	0.93 ***	0.60 ***	0.48 ***	0.44 ***	0.60 ***	−0.37 ***	41.10 (8.40)
Total score			−	0.63 ***	0.50 ***	0.46 ***	0.46 ***	−0.35 ***	83.91 (16.38)
Health anxiety									
Illness likelihood				−	0.70 ***	0.64 ***	0.95 ***	−0.42 ***	11.09 (4.34)
Illness severity					−	0.54 ***	0.82 ***	−0.37 ***	4.20 (1.97)
Body vigilance						−	0.68 ***	−0.35 ***	4.52 (1.82)
Total score							−	−0.44 ***	19.73 (7.31)
									Median (range)
Being close to people infected with COVID-19								-	2 (1–3)

Notes: Being close to people infected with COVID-19: 1 = yes + close contact; 2 = yes + no close contact; 3 = no. *** = *p* < 0.001.

**Table 3 ijerph-18-03094-t003:** Descriptive statistical indices of state trait anxiety and health anxiety, separately for being close to people infected with COVID-19 (yes + close contact; yes + no contact; no) and separately for working frontline (yes; no).

Dependent Variables	Being Close to People Infected with COVID-19					
	Yes + Close Contact		Yes + No Close Contact		No	
	Working Frontline		Working Frontline		Working Frontline	
	Yes	No	Yes	No	Yes	No
N	12	33	35	69	0	19
	M (SD)	M (SD)	M (SD)	M (SD)	M (SD)	M (SD)
State anxiety	43.25 (9.29)	48.21 (9.04)	40.74 (8.68)	42.45 (8.52)	0	38.26 (8.50)
Trait anxiety	40.42 (9.85)	47.82 (6.65)	36.83 (7.80)	41.55 (7.62)	0	36.11 (6.10)
Total score	83.67 (18.07)	96.03 (14.19)	77.57 (15.40)	84.00 (15.07	0	74.73 (13.53)
Health anxiety	22.08 (8.90)	24.82 (7.36)	19.94 (7.22)	18.57 (5.74)	0	13.21 (5.16)

**Table 4 ijerph-18-03094-t004:** Inferential statistical indices for state and trait anxiety and health anxiety, with the factors being close to people infected with COVID-19 and frontline working.

	Factor: Being Close to Infected People		Factor: Frontline Working		Interaction	
Degrees of freedom	(2, 163)		(1, 163)		(2, 163)	
	F	$\eta_{p}^{2}$[ES]	F	$\eta_{p}^{2}$[ES]	F	$\eta_{p}^{2}$[ES]
State anxiety	7.47 ***	0.084 [M]	3.74	0.022 [S]	0.89	0.005 [T]
Trait anxiety	14.66 ***	0.152 [L]	16.67 ***	0.093 [M]	0.81	0.005 [T]
Total score	12.21 ***	0.130 [M]	9.97 **	0.058 [S]	0.99	0.006 [T]
Health anxiety	16.93 ***	0.172 [L]	0.60	0.002 [T]	2.48	0.015 [T]

Notes: ** = *p* < 0.01; *** = *p* < 0.001. [ES] = effect size; T = trivial effect size; S = small effect size; M = medium effect size; L = large effect size.

**Table 5 ijerph-18-03094-t005:** Multiple linear regression with health anxiety as outcome variable, and state anxiety, trait anxiety, age, gender, working frontline, current job status, and being in contact with people infected with COVID-19 as predictors.

Dimension	Variables	Coefficient	Standard Error	Coefficient β	t	*p*	R	R^2^	Durbin–Watson
Health anxiety	Intercept	7.024	3.196	−	2.198	0.029	0.689	0.469	1.57
	State anxiety	0.209	0.070	0.260	2.993	0.003			
	Trait anxiety	0.315	0.08	0.363	3.898	0.000			
	Close to infected people ^a^	−2.573	0.766	−0.211	−3.358	0.001			
	Frontline ^b^	−2.58	0.989	−0.159	−2.612	0.013			
	Excluded variables: age, gender, current job status; all ts < 1.0; p’s > 0.30								

Notes: ^a^ Close to infected people: 1 = no; 2 = yes, but no direct contact; 3 = yes and direct contact. ^b^ Frontline: 1 = yes; 2 = no.

## Data Availability

The data presented in this study are under ownership of the funding institute (ARAKMU) and would not be available.

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
