# Peer review of "Sources of Health Anxiety for Hospital Staff Working during the Covid-19 Pandemic"

_ijerph, 2021, doi:10.3390/ijerph18063094_

Round 1
Reviewer 1 Report
Dear Editor,
Dear Authors,
I received the article, which I read with great interest. I would like to offer some ideas for a revision of the manuscript hoping to be helpful in refining the quality of the product.
1) The title should be made shorter and clearer. Minor details can be described more in the abstract instead of the title.
2) The theoretical background is poor. Association between constructs is reported, providing sufficient national and international literature review. However, I would like to read more about the theoretical aspects underlying the relationship between constructs.
3) The relationship between health anxiety and traumatic aspects in exposure to patients with COVID-19 has not been thoroughly evaluated, and I am sorry. Perhaps this is not the focus of your research work, however I would make a mention along the manuscript.
4) I would like to read a more thorough description regarding procedure and sample recruitment.
5) data referring to the association between trait anxiety and working frontline is interesting, but should theoretically be discussed in more depth in the discussion.
6) The limits section should be totally revised and deepened. Some possible limitations seem to be overlooked, such as the cross-sectional nature of the study and some aspects related to the use of self-report instruments. In addition, the first point discussed in the limitations does not sound appropriate in a scientific context, even more so when there are already systematic reviews or meta analysis.
7) Conclusions should better focus on the results obtained from the study and the goals that the scholars set. In addition, directions for future research, in the national and international context, should be explored. Finally, the authors should devote more space to discussing the practical implications of their study and possible intervention strategies.
Author Response
Dear Reviewer,
Thank you for all your kind efforts Please find the detailed point-by-point-response attached as a separate file.

Reviewer 2 Report
I suggest using always thousand separators as in this example: 16,291 and not 16’291 (unusual in my experience)
Table 4, page 6
I believe it would be better explaining in the Notes of the Table 4 the meaning of the symbols F and [ES]. Even in the Methods, Statistical Analysis, I was not able to find it.
Table 5, page 7
I suggest to mention the Durbin-Watson test in the Methods, Statistical Analysis.
Author Response

(The authors gave the same response as above.)

Reviewer 3 Report
The article is well-written and provides a sound scientific report of the study, which is of current interest to the scientific community. Statistics and the interpretation of the results are appropriate and demonstrate a careful approach to the data.
I attach the pdf of the paper with comments - some corrections to English language are suggested as indicated.

Author Response

(The authors gave the same response as above.)

Reviewer 4 Report
This paper is of general interest and provides study findings in the field of COVID-19 and related mental health relations. The quality of presentation is moderate. There are several limitations within the study. The originality and novelty of this study is limited, IJERPH published a very similar study some weeks ago. The interest to readers is moderate. The topic has been studied extensively in the last month.
See:
Investigating the Psychological Impact of COVID-19 among Healthcare Workers: A Meta-Analysis.
Int J Environ Res Public Health. 2020 Dec 5;17(23):9096. doi: 10.3390/ijerph17239096.Psychological Adjustment of Healthcare Workers in Italy during the COVID-19 Pandemic: Differences in Stress, Anxiety, Depression, Burnout, Secondary Trauma, and Compassion Satisfaction between Frontline and Non-Frontline Professionals.
Int J Environ Res Public Health. 2020 Nov 12;17(22):8358. doi: 10.3390/ijerph17228358.
Introduction: A theoretical background that gives a fundamental overview on relevant theories and studies is missing. The background, theory and relevant studies for this research must be presented concisely. The background information the authors described in the introduction section should provide the reader with any additional context or explanation needed to understand the results.
Methodology: Data presentation is not sufficient.
The authors should explain the techniques they you used to "clean" the data set.
Results: It would be helpful to include an introductory context for understanding the results by restating the research problem underpinning this study. This is useful in re-orientating the reader's focus back to the research after reading the literature review and your explanation of the methods of data gathering and analysis. The most important findings the authors want readers to remember should be highlighted (as a transition to the discussion section).
Discussion: The authors should organize the discussion from the general to the specific, linking their findings to the literature, then to theory, then to practice. The discussion section should relate the study findings to those found in other studies, particularly if questions raised from prior studies served as the motivation for this study. Systematically explain the underlying meaning of the findings.
Author Response

(The authors gave the same response as above.)

Reviewer 5 Report
The revised manuscript is an interesting work on the factors that have influenced the health anxiety of hospital staff members during the COVID-19 pandemic. The issue – Health anxiety, is significant, and a possible vulnerability factor because research has consistently demonstrated that individuals who score high on this construct tend to react to pandemic stress and health-related problems with intense negative affect.
The scientific structure presented is correct, but the sample used is small to get relevant data. Almost all different sections of the manuscript are adequately presented and meet the characteristics that any scientific work should have. The results respond to the objectives pursued, and the method used in the one we consider relevant.
However, there are different issues that we believe should be significantly improved. The main ones are listed below:
- Title: Need to revise the title; it is not concise, nor clear regarding the object of study. Concise and informative. Titles are often used in information-retrieval systems.
- Abstract: The abstract should be more concise. Abstract has multiple repetitions and non-relevant elements. The aims should be clarified. It could be: The aim of this study was to evaluate hospital staff members in terms of the relative contributions of sociodemographic and trait and state-anxiety variables on health anxiety.
- The introduction does not clearly state the problem being investigated, does not summarize relevant research to provide context, and does not explain other authors' findings. The conceptual framework needs a great deal of clarification. Part of the introduction needs to be adjusted and more concise. It will be better to add the literature review about sociodemographic factors related to health anxiety. More abundant references of current works dealing with the subject of this study need to be commented on in the introduction. The prevalence of health anxiety in health workers outside the current COVID-19 situation must be reviewed and to be compared against empirical findings in the actual situation.
Health anxiety is one of the most common types of anxiety. It is important to explore this construct in-depth. “The most important factor leading to health anxiety in health personnel, especially nurses, amid the COVID-19 pandemic is the high rate of healthcare worker infection and mortality”*. In this sense, the prevalence and incidence of COVID-19 in the study during the period of study should be discussed.
*References:
https://onlinelibrary.wiley.com/doi/pdf/10.1111/inm.12800?casa_token=QT9DZcmlNPEAAAAA:ZbvwYnHDe6C-9W2yPLl2Myan6qwPk5YSXYdYt7iOBrH0WwXdVqY3H3VwtyQtrVnaiXhoKSPoe-SqxZCJ
Health Anxiety, Perceived Stress, and Coping Styles in the Shadow of the COVID-19https://assets.researchsquare.com/files/rs-142530/v1/d80fc463-0a5d-4efd-b47d-2db88ca06748.pdf
Methods: is important to explain in greater detail the reason for the instruments chosen for the collection of information. Need to better describe the instruments: indicating values, cut-off, and the alphas obtained in the present work. How did you collect the data: face-to-face; online? How did the authors ensure that a representative sample of members of that organization took part? It would be convenient to describe the data collection process a little more. Are published specific normative STAI -S and STAI -T values for hospital staff? The values are available for any groups?
Discussion: Regarding the discussion, as in the introduction, a greater number of references would be desirable to provide greater consistency to said section and thus ensure that the results obtained can be much more consistent. Also explain in more detail in the discussions about health anxiety, depth in its causes, and especially, how to manage it proactively. Besides, the contextual factors that can influence this health anxiety should be presented. For example, the availability of personal protective equipment could influence health anxiety.
The limitations section needs to be rewritten. This study used a cross-sectional design, which does not allow for causal inferences.
References: the authors do not select appropriate and sufficient material to cite. The identification of references does not follow the standards (numbering) of the journal.
I wish the authors the best.
Author Response

(The authors gave the same response as above.)

Round 2
Reviewer 1 Report
.
Author Response
Again, we thank Reviewer #1 for the care devoted to further evaluate the revised manuscript. Please find the detailed point-by-point-response attached as a separate file.
Sincerely,

Reviewer 4 Report
The manuscript has been sufficiently revised. Many suggestions have already been implemented.
The connection between research questions and theory is still missing. The research questions are rather imprecise formulated.
The linking of the study results to the research question and to the theoretical background must be explicitly stated.
Author Response
Again, we thank Reviewer #4 for the care devoted to the revised manuscript. Please find the detailed point-by-point-response attached as a separate file.
Sincerely,

Reviewer 5 Report
Dear Editor,
Dear authours,
I am satisfied with the responses of the authors to my comments.
My decision is to accept the manuscript in the current version.
I thank the editors for the opportunity to review this manuscript and encourage the authors to continue working on this interesting line of research.
Best regards.
Author Response
Again, we thank Reviewer #5 for thoroughly review the revised manuscript.
